# Mechanoresponsive Material of AIE-Active 1,4-Dihydropyrrolo[3,2-b]pyrrole Luminophores Bearing Tetraphenylethylene Group with Rewritable Data Storage

**DOI:** 10.3390/molecules23123255

**Published:** 2018-12-10

**Authors:** Yuqing Ma, Yuyang Zhang, Lin Kong, Jiaxiang Yang

**Affiliations:** Anhui Province Key Laboratory of chemistry for Inorganic/Organic Hybrid Functional Materials, College of Chemistry & Chemical Engineering, Anhui University, Hefei 230601, China; 13053097532@163.com (Y.M.); zhangyy1003@163.com (Y.Z.); kong_lin2009@126.com (L.K.)

**Keywords:** 1,4-dihydropyrrolo[3,2-*b*]pyrrole derivative, aggregation-induced emission (AIE), reversible mechanochromism behavior, rewritable data storage

## Abstract

A new tetraphenylethylene (TPE) functionalized 1,4-dihydropyrrolo[3,2-*b*]pyrrole derivative (**APPTPECN**) was synthesized with obvious aggregation-induced emission (AIE) active by simple synthetic method. **APPTPECN** exhibited reversible mechanofluorochromic (MFC) behavior. The powder X-ray diffraction (PXRD) and scanning electron microscopy (SEM) investigations exhibited that the MFC nature is originated through a conversion from the microcrystalline to amorphous phase under the stimulus of external force. The results obtained would be of major help in understanding the MFC mechanism and designing new MFC materials. Compound **APPTPECN** has the potential possibility to employ in rewritable data storage and is of assistance in the rational design of smart luminescent materials.

## 1. Introduction

Fluorescent mechanofluorochromic (MFC) is a phenomenon where solid and liquid crystalline materials change their photoluminescence properties upon applying mechanical stimulation, such as grinding, ball-milling, and crushing [1,2,3,4,5,6,7]. Fluorescence emission and color change could often recover by another stimulation or more, such as heating, organic solvent vapor, and light [8,9,10,11,12,13,14,15,16]. Mechanofluorochromic materials as a kind of “smart material’’ have its potential applications in mechanosensors, fluorescence switches, and data storage so on [17,18,19,20,21,22,23,24,25,26]. Recently, rationally controlling the design of molecular mechanofluorochromic behavior is still a huge challenging, two main problems restrict the development of this kind of materials: the lack of deep understanding for the structure-property relationship of such materials and the fluorescent quenching effect in the solid state [27,28,29,30,31]. It is exciting that a major breakthrough came from an unusual aggregation-induced emission (AIE) phenomenon in the study of 1-methyl-1,2,3,4,5-pentaphenylsilole [32]. Since then, AIE concept has been extensively used to design efficient solid-state fluorescent materials and MFC materials because of highly twisted backbone structures bearing rotatable units [33,34,35]. Also, the twisted molecular conformations which not only enable them to emit strong fluorescence in the aggregated state by weakening intermolecular close stacking and intense π-π interactions, but also easily lead to the formation of MFC properties by changing the molecular packing modes upon pressure [36,37,38]. Herein, the synthesis of novel effective solid-state luminogens is still a hot topic for the researchers.

In order to obtain excellent MFC materials with strong fluorescence emission in the solid state and aggregated state, an artful design of the molecule was important for tuning molecular MFC properties. Typically, 1,4-dihydropyrrolo[3,2-*b*]pyrroles (PPs), a class of 10 π-electron aromatic dihydroheteropentalenes, were discovered by Hemetsberger and Knittel in 1972 [39]. Importantly, Gryko and co-workers developed a one-step synthetic method to acquire tetraaryl-pyrrolo[3,2-*b*]pyrroles in 2013 [40]. Moreover, these compounds have received increasing attention because PPs have a lot of advantages including high thermal, photostability, good absorptivity, bright fluorescence and high quantum yields [41,42,43]. Meanwhile, molecules and the modifications of TPE have been proved an efficient way to construct new aggregation-induced emission (AIE) luminogens with high solid-state efficiency [44,45,46,47,48,49]. 

Based on the above considerations, we present a simple method to discover MFC compound of novel 1,4-dihydropyrrolo[3,2-*b*]pyrroles derivative containing TPE unit. We have been trying our best knowledge to grow crystal to explain the phenomenon why the material fluorescence enhanced after grinding, unfortunately, we still unable to successfully cultivate crystal. **APPTPECN** showed strong AIE property. Also, the obtained **APPTPECN** exhibited reversible MFC behaviors. It was shown that interconversion between the microcrystalline and amorphous phase, and the MFC mechanism was proved by PXRD and SEM.

## 2. Results and Discussion

### Synthesis of the **APPCN** and **APPTPECN**

The synthesis of **APPCN** was carried out using one-step method and compound **APPTPECN** was synthesized using a classical Knoevenagel condensation reaction. The chemical structures and synthetic route of the compounds are shown in Scheme 1. The detailed procedure is listed as follows.

Considering the miscibility of DMF and H_2_O, for the purpose of checking whether **APPTPECN** is AIE-active, we used a standard approach to investigate UV-vis absorption and FL spectra. Concentrations of **APPTPECN** were kept at 1.0 × 10^−5^ mol L^−1^ in DMF and water mixture solution. For UV-vis absorption spectra, obviously, with the increasing water fraction, the solubility of the compound gradually decreased and the molecules gradually shifted from the free state to the aggregation state, forming the molecular aggregated particles. Meanwhile, the level-off tail suggested that the aggregates have been formed in mixtures (Figure 1a). DLS experiment showed the formation of aggregates with average hydrodynamic diameter of 265 nm (polydispersity index = 0.342) (Appendix A). For FL spectra, **APPTPECN** showed very weak visible light excited by 365 nm UV light in pure DMF because of the intramolecular rotation of tetraphenylethylene core. FL intensity remarkably increases from 30% to 99% water fraction and reaches its peak at 90% (Figure 1b–d). Also, there is a significant red-shift of **APPTPECN** fluorescence wavelength. The bright yellow-green light emitting of **APPTPECN** (located at 532 nm) was observed for the aggregates, which was about 30 times higher than that in the pure DMF, likely due to the intramolecular rotation restriction of tetraphenylethene unit [32,50]. These results indicated a typical AIE activity that the non-radiative processes were restrained because of the restriction of intramolecular rotation and intramolecular interaction [51,52,53,54].

FL spectra was utilized to investigate the MFC behavior of **APPTPECN** in the solid state. As shown in Figure 2a, The pristine solid **APPTPECN** emitted green fluorescence in 512 nm (Φ_F_ = 14%, τ < 0.1 ns, Appendix A). Interestingly, **APPTPECN** exhibited an obvious red-shift upon grinding (34 nm) yellow emission (λem = 546 nm, Φ_F_ = 22%) with a long lifetime (τ = 1.15 ns, Appendix A). Higher Φ_F_ values indicated a higher average excited state lifetime compared to the pristine sample. This result is worth noting that only a few materials have been reported to display higher fluorescence quantum yield for the pristine sample than that for the grinding sample in the field of smart materials [55,56,57,58]. Additionally, we take insight to determine the self-reversible mechanofluorochromic fluorescence conversion by alternating grinding and EtOH fuming, the fluorescence emission and colour of grinding powder could be nearly converted numerous times between the green and yellow emissions (Figure 2b). It was possible that an external mechanical force resulted in a change in the molecular morphology, indicative an outstanding reversibility of the MFC process. Owing to the excellent mechanofluorochromic properties of the smart fluorescent materials. We further explore the practical application, a simple and portable ink-free rewritable paper were conducted on Figure 2c, the pristine sample was dissolved in CH_2_Cl_2_ (DCM), then a Whatman filter paper was immersed into the solvent for approximately 30 min. Afterwards, Whatman filter paper was taken out and dried at room temperature. Intriguingly, they were very clear and discernible under 365 nm irradiation and filter paper emitted green emission. When the **APPTPECN**-loaded paper was crushed, the fluorescence emission significantly red-shift. The letter “light” can be erased after the “paper” was exposed to DCM vapor. Furthermore, a Chinese letter “light” was easily written in the Whatman filter paper again. As a result, the material has great potential for application in ink-free rewritable paper.

The excellent rewritable paper further promotes us explore the application of rewritable data storage, as shown in Figure 3, compared to green background, the preparation of Whatman filter paper was crushed and appear yellow ‘H’ letter. Then, a green background is easily also converted into a yellow background with crushing. Next, ‘H’ and ‘Y’ letters were written using a specially made ‘pen’ with DCM vapor as ‘ink’, which exhibiting green fluorescence emission due to the grinding sample back to the original state after DCM treatment. Last, exposure of the Whatman filter paper to DCM vapor can readily erase the fluorescent letters and return to original green fluorescence background. Herein, compound **APPTPECN** is employed in rewritable data storage and is of assistance in the rational design of smart luminescent materials.

To further understand and evaluate the origin and MFC properties in the powder state, PXRD was investigated for the sample before and after grinding. As shown in Figure 4, the diffraction curves of the pristine sample revealed many sharp and intense reflections, indicative of the well-ordered crystalline structures. In contrast, after grinding, the diffraction peaks nearly disappeared and only some diffuse, broad and weak band were observed, exhibiting disordered molecular packing or amorphous states. The process implied that grinding led to a microcrystalline to amorphous phase conversion of solids. After exposing the grinding sample to EtOH vapor, the sharp and intense diffraction peaks reappeared. Therefore, it was manifested **APPTPECN** has the reversible mechanofluorochromic behavior.

The aggregation morphology of **APPTPECN** before and after grinding were investigated by scanning electron microscopy (SEM) investigations. As performed in Figure 4, the **APPTPECN** exhibited an analogous “magnesium” structure (Figure 5a), whereas the irregular and rough aggregates were observed for the grinding **APPTPECN** (Figure 5b). When the grinding sample was fumed by EtOH, it essentially reverted to its original microcrystalline state (Figure 5c). It is further proved that a microcrystalline to amorphous phase conversion of solid after grinding.

TG analysis curves revealed that the degradation temperature (Td) of 5% weight loss of **APPTPECN** before and after grinding are 438 °C, 440 °C, respectively (Figure 6a). This result revealed that **APPTPECN** has an excellent thermal stability. In addition, DSC thermograms demonstrated that a new exothermic peak at 159 °C was observed for grinding sample. This exothermic peak could be attributed to the cold-crystallization transition of the amorphous **APPTPECN** caused by mechanical grinding (Figure 6b). Hence, PXRD, SEM and DSC experiments verified that phase transformation between the microcrystalline and amorphous states should be responsible for the MFC behavior.

## 3. Materials and Methods

### 3.1. Measurements and Instrument

All of the chemicals and solvents were purchased from Darui of Shanghai without further purification. ^1^H NMR (400 and 600 MHz) spectra was registered in a Bruker Avance with CDCl_3_ as the solvent and tetramethylsilane (TMS) as the internal standard. The UV-vis absorption spectra was measured in a TU-1901 spectrometer (Purkinje General Instrument Co., Ltd. Beijing, China). The fluorescence spectra were recorded on a Hitachi FL-7000 (Shanghai, China). The absolute solid-state fluorescence quantum yield in the solid state was conducted on a HORIBA FluoroMax-4 spectrofluorometer (Irvine, CA, USA) with an integrating sphere. The powder X-ray diffraction (PXRD) patterns were performed on an MXP18AHF (Hefei, China). The thermogravimetry (TG) analysis using TGA5500 and differential scanning calorimetry (DSC) was measured at a heating rate of 10 °C·min^−1^ under a N_2_ atmosphere using DSC apparatus (METTLER82le/400 (Zurich, Switzerland)).

### 3.2. Synthesis

#### 3.2.1. The Generally Synthetic Procedure of Compound 2,6-di(4-Cyanomethylphenyl)-1,5-di(4-cyanophenyl)-1,4-dihydropyrrolo[3,2-*b*]pyrrole (**APPCN**)

4-Aminobenzyl cyanide (1.0 g, 7.6 mmol), *p*-cyanobenzaldehyde (886 mg, 7.6 mmol) and *p*-toluenesulfonic acid (45 mg, 757 μmol) were dissolve in 10 mL glacial acetic acid, the mixture were heated to 90 °C, keeping the temperature for 1 h, Then, followed by addition of 2,3-Butanedione (651 mg, 3.8 mmol), the resulting mixture was stirred at 90 °C for 3 h. The reaction was cooled to ambient temperature. Then, the crude product was purified by washing with ethyl acetate. Yellow solid, 700 mg. Yield 35%. ^1^H NMR (400 MHz, DMSO-*d*_6_): *δ* (ppm): 4.12 (s, 2H), 6.76 (s, 1H), 7.34 (d, *J* = 7.44 Hz, 2H), 7.38 (d, *J* = 7.44 Hz, 2H), 7.46 (d, *J* = 7.76 Hz, 2H), 7.73 (d, *J* = 7.48 Hz, 2H). FT-IR (KBr, cm^−1^): 3061, 2939, 2224, 1600, 1514, 1457, 1420, 1366, 1177, 1140, 843. MALDI-TOF: *m*/*z* = 539.02 ([M + H]^+^), calcd for C_36_H_22_N_6_^+^ = 539.12([M + H]^+^).

#### 3.2.2. The Generally Synthetic Procedure of Compound **APPTPECN**

A solution of 4-(1, 2, 2-triphenylvinyl)benzaldehyde (294 mg, 0.82 mmol) in EtOH-THF (10 mL, v/v = 1:1.5) was heated to reflux, then, **APPCN** (200 mg, 0.37 mmol) and *t*-BuOK (167 mg, 1.5 mmol) were added into solution, stirred 5 h. Then, The reaction was cooled to room temperature and filtration. The target compound was purified by washing with heated EtOH. Yellow solid, 272 mg, Yield: 60%. ^1^H NMR (600 MHz, CDCl_3_): *δ* (ppm): 6.54 (s, 1H), 7.04–7.14 (m, 17H), 7.31–7.32 (m, 4H), 7.43 (s, 1H), 7.54 (d, *J* = 7.62 Hz, 2H), 7.67–7.70 (m, 4H). ^13^C NMR (150 MHz, CDCl_3_): *δ* (ppm): 146.92, 143.27, 143.12, 142.57, 142.28, 139.94, 139.64, 137.22, 135.17, 133.02, 132.23, 132.01, 131.35, 131.33, 131.27, 128.83, 128.07, 127.91, 127.89, 127.70, 127.10, 126.94, 128.78, 125.54, 118.84, 117.91, 109.81, 109.38, 97.07, FT-IR (KBr, cm^−1^): 3077, 3045, 2219, 1594, 1511, 1440, 1389, 1168, 1072, 835. MALDI-TOF: *m*/*z* = 1223.30 ([M + H]^+^), calcd for C_86_H_60_N_6_^+^ = 1223.37 ([M + H]^+^).

## 4. Conclusions

In this work, a novel molecular design of 1,4-dihydropyrrolo[3,2-*b*]pyrrole derivative (**APPTPECN**) containing TPE was synthesized, The incorporation of TPE unit in the backbone endows the **APPTPECN** an obvious aggregation-induced emission (AIE) behaviour. Effective MFC behavior with the red-shifts of 34 nm in FL spectra upon grinding is observed and indicative of reversible performance. Moreover, compound **APPTPECN** has the potential possibility to employ rewritable data storage and is of assistance in the rational design of smart luminescent materials. The present study provides valuable information for designing new AIE MFC materials with desired properties.

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
