# Peer review of "Mechanoresponsive Material of AIE-Active 1,4-Dihydropyrrolo[3,2-b]pyrrole Luminophores Bearing Tetraphenylethylene Group with Rewritable Data Storage"

_molecules, 2018, doi:10.3390/molecules23123255_

Round 1

Reviewer 1 Report

It is an interesting manuscript in which the synthesis of 1,4-dihydropyrrolo[3,2-b]pyrrole luminophores and potential application is described. The manuscript presents valuable information related to design and explanation of the mechanism of action as well as practical use of AIE-active luminophores.

Both the description of the measurement methodology and the measurement methods used are correct. Interpretation of the results obtained also does not raise doubts.

However, the quality of the manuscript could be increased by placing omitted drawings (Figs 6a and b) and reference  titles: ref. 16 line 233.

Minor shortcoming:

line 49 instead of high it is better to write bright,

line 62 “knoevenagel” with a capital letter because it's a surname,

line 161 specify the type of the apparatus used for TG and DSC measurements,

line 306 correct “donoreacceptor”.

Author Response

Thank you for your kind comment, according your valuable advice, we have checked and corrected.

Reviewer 2 Report

In this contribution Ma and coworkers describe a new chromophore exhibiting AIE properties with potential applications relation of its  re-writable properties.

This paper is interesting but request significant style and english language correction (not listed here). The characterizations of the compounds are also not sufficient.

A list of points to correct is given below:

1- The term MFC is not defined in the abstract.

2- Contrary to the statement of the authors in the abstract, the chromophore is not employed in data security storage. This is only a potential application. Same remark for the conclusion.

3- In the introduction, please cite the following paper relation to AIE properties of PPs derivatives: Org. Lett. 2018, 20, 3183 and Mater Chem. Frontiers 2018, 2, 1175.

4- l 61: APPCN and APPTPECN should be in bold

5- Scheme 1: please add the yield of the reactions. For the second line, APPCN is missing as reactent.

6- Fig 1b, Fig1c and Fig 1d are not cited in the text of the manuscript.

7- The paragraph related with data security storage is not clear. I do not understand the link between what is shown and  data security storage.

8- APPCN is not clean according to the copy of its 1H NMR spectra. This compound should be purified again and the provided yield should be of the spectroscopically clean compound.

9- Integration number of 1H NMR description should be doubled.

10- APPCN and APPTPECN should be characterized by 13C NMR and HRMS (or elemental analysis)

11- l178 ...washing with EtOH.

12- l183: "an artistic molecular desgin" is not a scientific term.

Author Response

Thank you for your valuable advice and careful comment, according your suggestion, we have checked and corrected point by ponit.

Round 2

Reviewer 2 Report

The manuscript has been improved but magor correction are still required:

1- Scheme 1: please add the yield of each reaction

2- Fig 2a: the spectra ere normalized: no unit should be provided for y-axis legend (please remove a.u.)

3- Low resolution MS is not enough: high resolution mass spectroscopy (HRMS) results with four decimal and difference between experimental and calculated value below 5 ppm are required.

4- for maldi results if [M+H]+ is observed, the calculated value should be for [M+H]+ instead of [M]+

5- APPCN has not been purified, the same spectrum with another scale has been provided.

Author Response

Thank you for your comments, we have carefully checked and corrected in revised manuscript.
